# National genomic surveillance integrating standardized quantitative susceptibility testing clarifies antimicrobial resistance in Enterobacterales

Shizuo Kayama [1,2] ✉, Koji Yahara [1,2] ✉, Yo Sugawara [1] ✉, Sayoko Kawakami[1], Kohei Kondo[1], Hui Zuo[1], Shoko Kutsuno[1], Norikazu Kitamura[1], Aki Hirabayashi[1], Toshiki Kajihara[1], Hitomi Kurosu[1], Liansheng Yu[1], Masato Suzuki[1], Junzo Hisatsune[1] & Motoyuki Sugai[1] ✉

Antimicrobial resistance is a global health concern; Enterobacterales resistant to third-generation cephalosporins (3GCs) and carbapenems are of the highest priority. Here, we conducted genome sequencing and standardized quantitative antimicrobial susceptibility testing of 4,195 isolates of *Escherichia coli* and *Klebsiella pneumoniae* resistant to 3GCs and Enterobacterales with reduced meropenem susceptibility collected across Japan. Our analyses provided a complete classification of 3GC resistance mechanisms. Analyses with complete reference plasmids revealed that among the $bla_{CTX-M}$ extended-spectrum β-lactamase genes, $bla_{CTX-M-8}$ was typically encoded in highly similar plasmids. The two major AmpC β-lactamase genes were $bla_{CMY-2}$ and $bla_{DHA-1}$. Long-read sequencing of representative plasmids revealed that approximately 60% and 40% of $bla_{CMY-2}$ and $bla_{DHA-1}$ were encoded by such plasmids, respectively. Our analyses identified strains positive for carbapenemase genes but phenotypically susceptible to carbapenems and undetectable by standard antimicrobial susceptibility testing. Systematic long-read sequencing enabled reconstruction of 183 complete plasmid sequences encoding three major carbapenemase genes and elucidation of their geographical distribution stratified by replicon types and species carrying the plasmids and potential plasmid transfer events. Overall, we provide a blueprint for a national genomic surveillance study that integrates standardized quantitative antimicrobial susceptibility testing and characterizes resistance determinants.

Antimicrobial resistance (AMR) is one of the greatest threats to human health and, thus, requires effective surveillance[1]. In particular, the resistance of Enterobacterales to third-generation cephalosporins (3GCs) is responsible for the largest number of AMR-associated deaths, at least in the European Union and the European economic area[2]. Meanwhile, globally, 3GC-resistant *Escherichia coli* and *Klebsiella pneumoniae* were estimated to cause between 50,000 and 100,000 resistance-attributable deaths in 2019, respectively[3]. In addition, the

[1]Antimicrobial Resistance Research Center, National Institute of Infectious Diseases, Tokyo, Japan. [2]These authors contributed equally: Shizuo Kayama, Koji Yahara. ✉e-mail: kayama@niid.go.jp; k-yahara@niid.go.jp; suga-yo@niid.go.jp; sugai@niid.go.jp

resistance of any pathogen to carbapenems—a class of "last-resort" antimicrobials[4]—is considered a serious global public health threat, responsible for an estimated 243,000 deaths in 2019[2].

Although surveillance of AMR pathogens is typically conducted using routine phenotypic antimicrobial susceptibility testing, genome sequencing can complement these methods by providing information regarding resistance determinants and associated mechanisms[5]. In fact, national surveillance of Enterobacterales (particularly *E. coli* and *K. pneumoniae*) resistant to 3GCs and/or carbapenems complemented by genome sequencing was recently pioneered in the Philippines via a binational collaboration[6], which successfully advanced the current understanding of AMR determinants and vehicles underlying the expansion of specific resistance phenotypes.

However, the study conducted in the Philippines was limited by inconsistencies in the antimicrobial panels that were tested by the sentinel sites, thus highlighting the importance of standardized phenotypic antimicrobial susceptibility testing[6]. In addition, the project sequenced Enterobacterales isolates non-susceptible to carbapenems and/or 3GCs, although carbapenemase genes can be found in strains with a minimum inhibitory concentration (MIC) for meropenem below or equal to the clinical susceptibility breakpoint[7]. The European Committee on Antimicrobial Susceptibility Testing (EUCAST) defines the epidemiological cutoff for screening carbapenemase as a meropenem MIC $\geq 0.25\,\mu g/mL$. Furthermore, the previous study in the Philippines only examined extended-spectrum β-lactamase (ESBL) production, without assessing other determinants of resistance (e.g., AmpC β-lactamase[8]) to 3GCs.

Therefore, in this study, we aimed to conduct a national genomic surveillance of Enterobacterales resistant to 3GCs and those satisfying the epidemiological cutoff for screening carbapenemase; specifically, 5143 isolates were sequenced and antimicrobial susceptibility testing was further conducted for 4195 isolates using the same panel, thus enabling quantitative (i.e., wide-range) re-measurement of MICs. By combining the national genome sequencing data with the antimicrobial susceptibility testing results, we were able to provide a complete overview of the 3GC resistance mechanisms. This study could further provide a detailed snapshot of the current national distribution of 3GCs and carbapenem resistance determinants, as well as the plasmids encoding these determinants and reconstructed using long-read sequencing.

## Results

### Complete classification of 3GC resistance mechanisms

Among the 4195 strains that were collected across Japan and subjected to genome sequencing and re-measurement of MICs at the National Institute of Infectious Diseases, 97.4% ($N = 4088$) were resistant to one 3GC: cefotaxime [CTX], ceftazidime [CAZ], or ceftriaxone [CTRX] ((1) in Fig. 1). Among the remaining 2.6% ($N = 107$), only three strains were resistant to another 3GC (cefpodoxime [CPDX]), and the remaining 104 susceptible strains likely reflected errors in antimicrobial susceptibility testing at the hospitals participating in the surveillance.

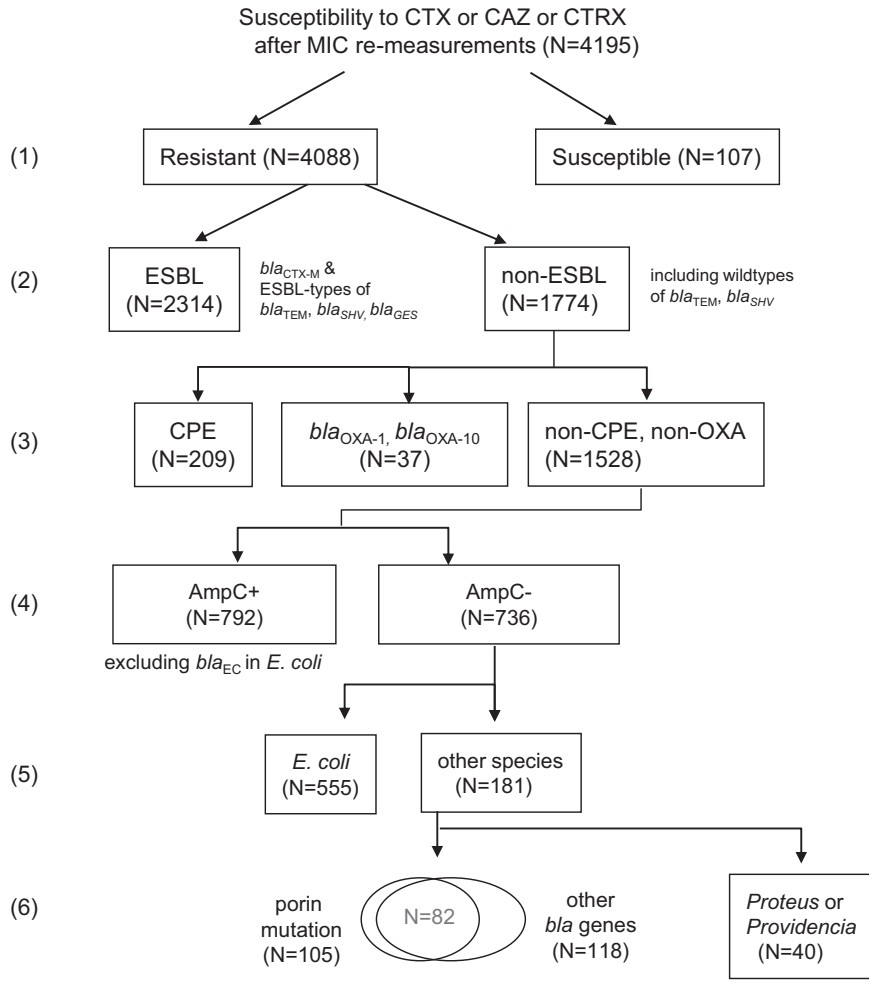

**Fig. 1 | Classification of 3GC resistance.** The 4195 isolates were classified in terms of (1) susceptibility to 3GC (i.e., CTX, CAZ, or CTRX), (2) presence or absence of an ESBL gene, (3) presence or absence of a carbapenemase gene or $bla_{OXA-1}$ or $bla_{OXA-10}$, (4) presence or absence of an AmpC β-lactamase gene, (5) *E. coli* or other species, (6) presence or absence of porin mutations and other β-lactamase genes. The numbers (1) to (6) are referred to in the main text.

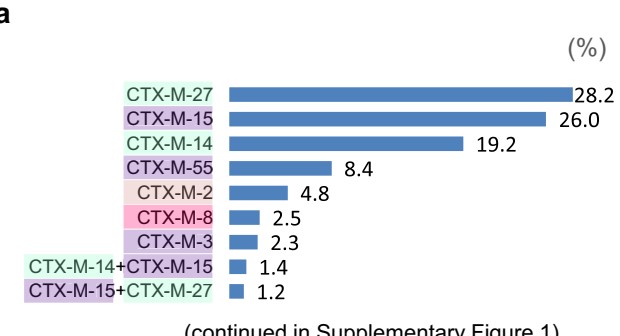

a

... (continued in Supplementary Figure 1)

among *E. coli* carrying at least a $bla_{CTX-M}$ gene (N=1932)

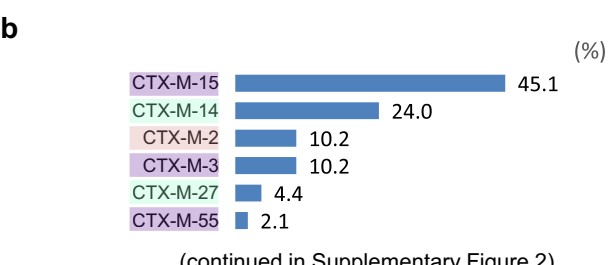

b

... (continued in Supplementary Figure 2)

among *K. pneumoniae* carrying at least a $bla_{CTX-M}$ gene (N=890)

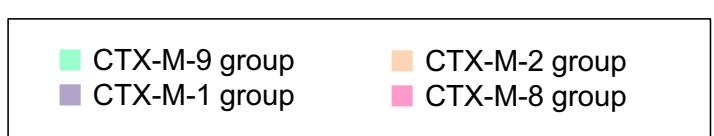

**Fig. 2 | Breakdown of major $bla_{CTX-M}$ genes in (a) *E. coli* and (b) *K. pneumoniae*.** The four colors on the left side of the bar plots represent four different $bla_{CTX-M}$ groups. Minor $bla_{CTX-M}$ genes with <1% frequency are abbreviated and shown in Supplementary Figs. 1 and 2.

The 4088 3GC-resistant strains were initially classified as ESBL producers ($N = 2314$) and non-ESBL producers ($N = 1774$) ((2) in Fig. 1); the former encode $bla_{CTX-M}$ or ESBL-types of $bla_{SHV}$ or $bla_{TEM}$ or $bla_{GES}$ (Supplementary Dataset 1). Moreover, the Kendall's coefficient of concordance (W) between carriage of an ESBL gene and 3GC resistance was 0.55 among the 4088 3GC-resistant and 107 3GC-susceptible strains. The non-ESBL strains ($N = 1774$) were classified as carbapenemase-producing Enterobacterales (CPE; $N = 209$), non-CPE strains encoding $bla_{OXA-1}$ or $bla_{OXA-10}$ ($N = 37$), and others ($N = 1528$; (3) in Fig. 1), which were further classified by the presence or absence of an AmpC β-lactamase gene ((4) in Fig. 1). Among the strains without the AmpC gene ($N = 736$), 75.3% were *E. coli* ((5) in Fig. 1), of which most (545 of 555) likely exhibited increased expression of the chromosomal AmpC ($bla_{EC}$) gene[9–11] as they were susceptible to cefepime (MIC values ≤ 2 μg/mL), which is minimally affected by high-level AmpC production[12]. In fact, a disk diffusion method using a subset of five *E. coli* strains with cefepime MIC ≤ 2 μg/mL (JBABADF-19-0021, JBCIAAH-19-0389, JBCIAEJ-19-0056, JBEAAEI-19-0005, JBBDAAF-19-0014) consistently confirmed the presence of AmpC enzyme produced at high levels. We also confirmed that the five *E. coli* strains had mutations in the promoter region of the chromosomal AmpC ($bla_{EC}$) gene, which can increase gene expression. Among the total 545 *E. coli* strains with cefepime MIC ≤ 2 μg/mL, 63% (341) carried

five types of mutations in the promoter region (C-11T, C-42T, G-15GG, T-14TGT, T-32A), as detected by AMRFinderPlus[13]. The proportion was significantly higher ($P = 0.008$, Fisher's exact test) than 20% (2 of 10) of *E. coli* strains with cefepime MIC > 2 μg/mL suggesting that AmpC enzyme is not produced at high levels. Meanwhile, the remaining 37% of 545 *E. coli* strains with cefepime MIC ≤ 2 μg/mL likely exhibited other mechanisms that were undetectable by AMRFinderPlus, for example the incorporation of IS10 and IS911 into the promoter region[14].

Among the remaining 181 strains of the other species (right of (5) in Fig. 1), 105 harbored mutations in the outer membrane porin genes and 118 harbored other β-lactamase genes ((6) in Fig. 1). After excluding 141 strains that harbored either of these genes, 40 strains of *Proteus* spp. or *Providencia* spp. remained, of which 38 showed MEPM MIC ≤ 0.5 μg/mL and 2 exhibited MEPM MIC ≥ 4 μg/mL. The determinants of MEPM resistance in these two strains are currently unknown.

**National distribution of $bla_{CTX-M}$ ESBL genes and plasmids**
In addition to the 4195 strains mentioned above, we obtained the genome sequences of 948 strains without re-measurement of MICs. At the multilocus sequence type (MLST) level, among the 5143 strains, 3159 were *E. coli* and 1240 were *K. pneumoniae*, the distribution of $bla_{CTX-M}$ genes for which is shown in Fig. 2 and Supplementary Fig. 1 and 2. The

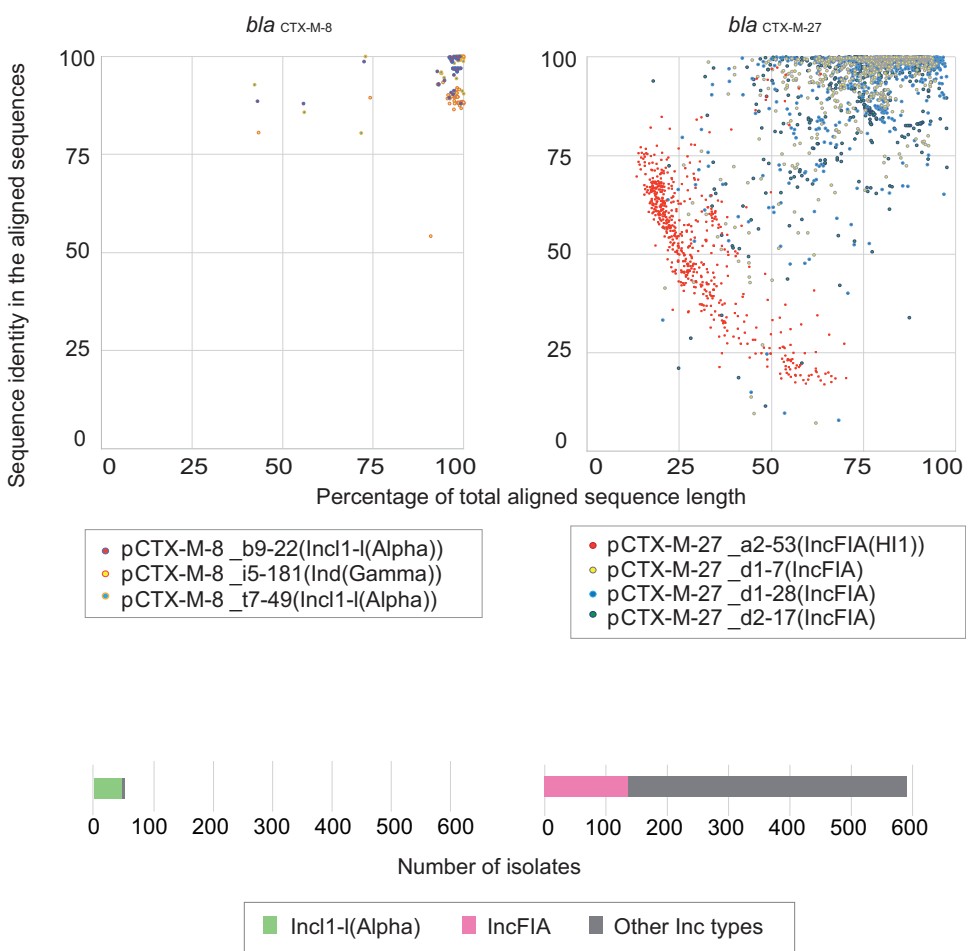

**Fig. 3 | Extent to which complete reference plasmid sequences were aligned to draft genomes including a single $bla_{CTX-M}$ gene.** Top: The x-axis indicates the percentage of total aligned sequence length; the y-axis indicates sequence identity in the aligned sequences. Colors represent different reference plasmids plotted for each isolate. The left panel uses the complete plasmid sequences carrying $bla_{CTX-M-8}$; the right panel uses panels carrying $bla_{CTX-M-27}$. The results for isolates carrying other $bla_{CTX-M}$ genes are shown in Supplementary Fig. 7. Bottom: Summary of the scatter plots at the top in terms of the proportion of strains satisfying the >90% total aligned sequence length and >90% identity in the aligned sequences against each reference complete plasmid sequence. Green and pink represent two distinct replicon types of the plasmids, while all the other replicon types are represented in gray. Source data are provided as a Source Data file.

$bla_{CTX-M}$ genes in *E. coli* accounting for >15% were $bla_{CTX-M-27}$ (28.2%) belonging to the CTX-M-9 group, $bla_{CTX-M-15}$ (26.0%) belonging to the CTX-M-1 group, and $bla_{CTX-M-14}$ (19.2%) belonging to the CTX-M-9 group, whereas those in *K. pneumoniae* were $bla_{CTX-M-15}$ (45.1%) and $bla_{CTX-M-14}$ (24.0%; Fig. 2). The most frequent sequence types (STs) accounting for >5% of *E. coli* were ST131 (46.8%), followed by ST73 (11.2%), and ST1193 (5.6%), whereas those in *K. pneumoniae* were ST307 (6.0%) and ST25 (5.3%; Supplementary Fig. 3). The geographical distributions of STs and $bla_{CTX-M}$ genes in *E. coli* are shown in Supplementary Fig. 4 and those in *K. pneumoniae* are shown in Supplementary Fig. 5.

$bla_{CTX-M}$ genes are typically encoded in a plasmid. For the 2850 strains encoding a single $bla_{CTX-M}$ gene, genome alignment between contigs of each strain and the complete reference plasmid sequences encoding the $bla_{CTX-M}$ genes with >2% frequency in *E. coli* (Fig. 2) revealed that the known three complete plasmid sequences encoding the $bla_{CTX-M-8}$ gene were aligned well with the genome sequences obtained in this study (Fig. 3, left). The percentage of total aligned sequence length (x-axis in Fig. 3) and identity in the aligned sequences (y-axis in Fig. 3) were relatively consistently >90% among the strains harboring a single $bla_{CTX-M-8}$ gene collected across the country. Hence, the $bla_{CTX-M-8}$ gene showed associations with plasmids that were highly similar to the reference complete plasmids, although this threshold cutoff does not rule out the potential for other ESBL-encoding gene genomic contexts.

Regarding the $bla_{CTX-M-27}$ gene, the same analysis revealed that one of the four complete reference plasmid sequences harboring the IncFIA replicon did not exhibit >90% total aligned sequence length or >90% shared identity in the aligned sequences (indicated in red to the right in Fig. 3), whereas the other three complete reference plasmid sequences partially satisfied the criteria (23.7% among the strains; bottom right in Fig. 3). These results suggested that 23.7% of the strains harboring a single $bla_{CTX-M-27}$ gene encoded it on a plasmid that was similar to one of the three complete reference plasmid sequences. Previous studies have reported that the $bla_{CTX-M-27}$ gene is associated with a large 131 kb transferable IncF1:A2:B20 plasmid. In fact, pairwise alignment between the 131 kb plasmid and the four complete reference plasmid sequences revealed that three sequences exhibited overall structural similarity (Supplementary Fig. 6).

Regarding other $bla_{CTX-M}$ genes, the proportion of strains satisfying the criteria of total sequence length aligned against >90% of the complete plasmid sequences and >90% identity in the aligned sequences notably reduced (Supplementary Fig. 7), suggesting that they are likely encoded in other unknown plasmids. For each $bla_{CTX-M}$ gene, including $bla_{CTX-M-8}$ and $bla_{CTX-M-27}$, the proportions of strains satisfying the criteria stratified by geographical region (southwest, west, middle, east, and north) are depicted in Supplementary Fig. 8. Excluding $bla_{CTX-M-8}$ and $bla_{CTX-M-2}$, the proportion of strains satisfying the criteria did not exceed 30% in any geographical region,

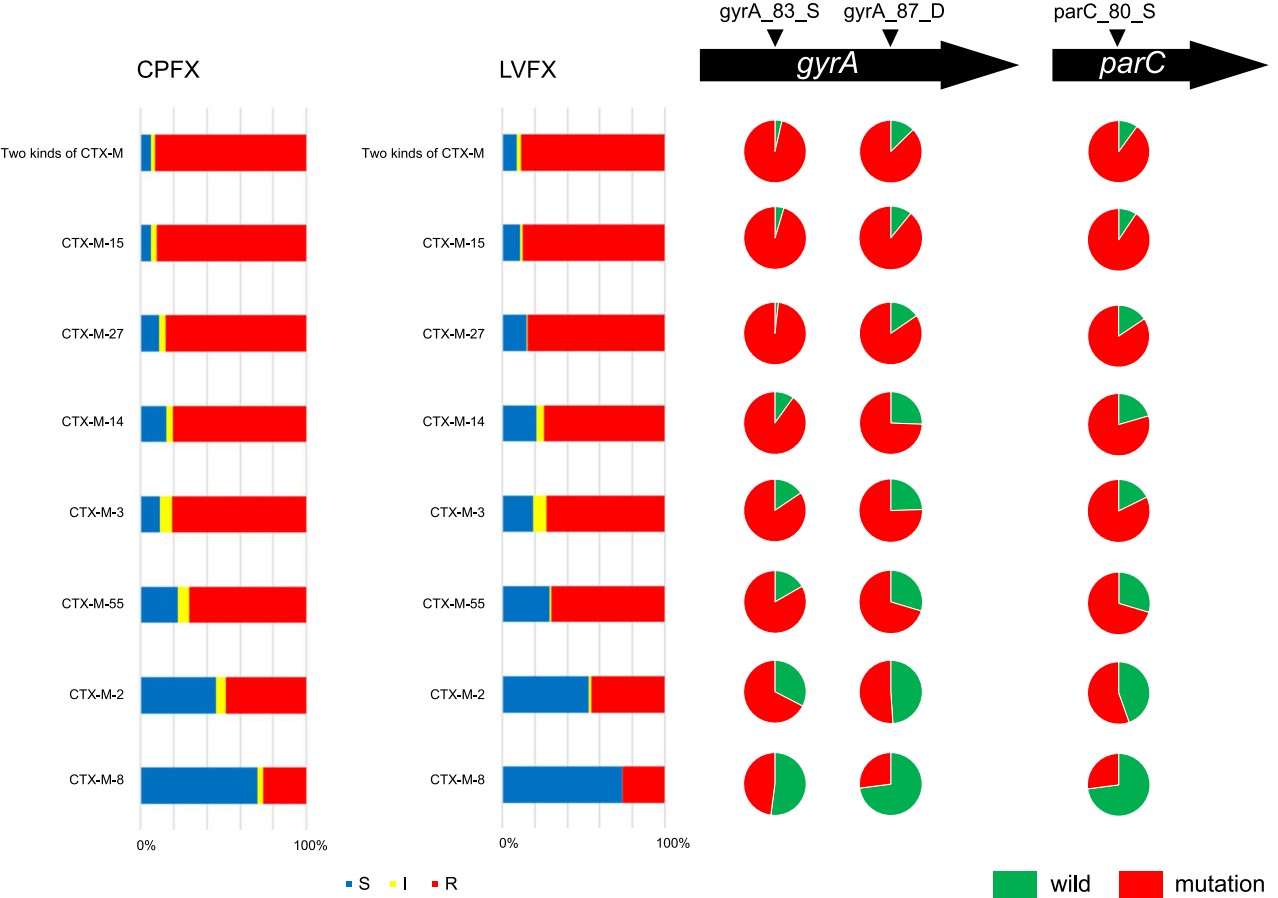

**Fig. 4 | Relationship between QRDR mutations and fluoroquinolone resistance stratified by *bla*CTX-M genes in *E. coli*.** Bar charts on the left indicate the antibiograms for two major fluoroquinolones (CPFX and LVFX). Pie charts on the right indicate the proportion of strains carrying each of the three QRDR mutations, including *gyrA* and *parC*. Blue, yellow, and red represent susceptibility, intermediate resistance, and resistance to the fluoroquinolones, respectively. Source data are provided as a Source Data file.

suggesting that other unknown plasmids are also distributed across the country.

## Notable antimicrobial susceptibility of strains harboring the ESBL genes

Certain ESBL strains are susceptible to piperacillin/tazobactam (PIPC/TAZ), which represents another potential treatment option[15]. Among the 2314 ESBL strains (Fig. 1), 1488 were *E. coli* and 727 were *K. pneumoniae* strains at the MLST level. Antimicrobial susceptibility profiles of the strains are depicted in Supplementary Fig. 9, indicating that the proportion of ESBL strains susceptible to PIPC/TAZ was 95.1% in *E. coli* and 82.8% in *K. pneumoniae*. Although a susceptibility of 95.1% is high, strains co-harboring ESBL and *bla*OXA-1 genes are reportedly associated with elevated PIPC/TAZ MICs and increased risk of mortality[16]; the proportion of co-harboring strains in our dataset was 17.8% in *K. pneumoniae* and 11.2% in *E. coli*, which was higher than the 2.9% (17/581 ESBL-producing *E. coli* strains) reported from seven facilities in a geographical region in Japan from 2001 to 2010[17]. These results suggest the risks of PIPC/TAZ as a treatment option for ESBL-producing *E. coli* and *K. pneumoniae* infections.

Fluoroquinolone-resistant *E. coli* is a priority pathogen that is subject to comprehensive AMR surveillance[18] and has been increasing in frequency in Japan. The relationship between the proportion of levofloxacin (LVFX) resistance and mutations in the quinolone resistance-determining regions (QRDRs) in *E. coli*, stratified by seven *bla*CTX-M genes, is shown in Fig. 4. Notably, strains harboring *bla*CTX-M-8

had significantly fewer nonsynonymous mutations in QRDRs (bottom right in Fig. 4) and thus had lower resistance to ciprofloxacin (CPFX) and levofloxacin (LVFX) (bottom left in Fig. 4) than other strains ($P < 10^{-9}$ for each of GyrA83, GyrA87, ParC80, resistance to CPFX and LVFX, chi-square test).

Regarding resistance to β-lactams, the susceptibility rates to ampicillin/sulbactam (ABPC/SBT) in *E. coli* were 73.4% and 18.2% among strains harboring *bla*CTX-M-27 and *bla*CTX-M-15 (after excluding those harboring carbapenemase genes, AmpC β-lactamase genes, and multiple *bla*CTX-M), respectively (Supplementary Fig. 10). A main factor underlying the large difference was the frequency of *bla*TEM-1 gene co-occurrence (Supplementary Dataset 2 and Supplementary Fig. 11), which is known to cause resistance to ABPC/SBT[19]. Kendall's W between the *bla*TEM-1 gene co-occurrence and ABPC/SBT resistant/intermediate/susceptible phenotype was 0.74 among strains harboring *bla*CTX-M-15 or *bla*CTX-M-27. However, the concordance was lower among strains harboring *bla*CTX-M-14 (Kendall's W = 0.57), suggesting the presence of another unknown factor. This is reflected in that 39% of strains non-susceptible to ABPC/SBT did not carry *bla*TEM-1 (top left in Supplementary Fig. 11).

In addition, strains harboring *bla*CTX-M-27, *bla*CTX-M-15 and *bla*CTX-M-55 tended to show lower susceptibility to CAZ and AZT (Supplementary Fig. 10, ≤33.0% for CAZ and ≤8.6% for AZT) than the other strains. These CTX-M enzymes carry a single amino acid substitution, Asp240-Gly, which is characteristic of the CTX-M-32 enzyme derived from CTX-M-1[20] and is responsible for high-level resistance to CAZ and AZT.

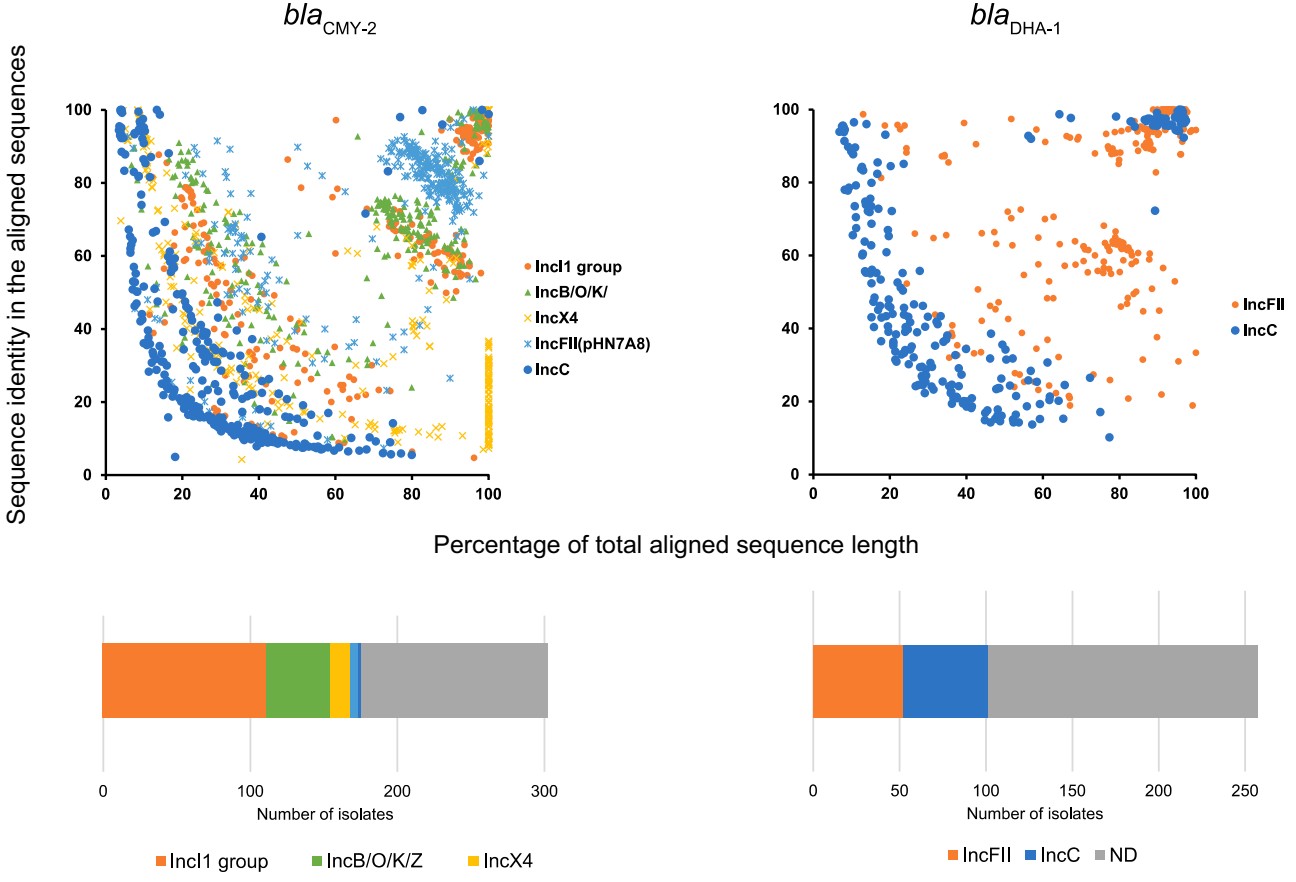

**Fig. 5 | Extent to which complete reference plasmid sequences could be aligned to draft genomes including a single AmpC beta-lactamase gene.** Top: The x-axis indicates the percentage of total aligned sequence length; the y-axis indicates sequence identity in the aligned sequences. Colors represent different reference plasmids plotted for each isolate. The left panel uses the complete plasmid sequences carrying $bla_{CMY-2}$; the right panel uses panels carrying $bla_{DHA-1}$. Bottom: Summary of the scatter plots at the top in terms of the proportion of strains satisfying the >90% total aligned sequence length and >90% identity in the aligned sequences against each reference complete plasmid sequence. Orange, green, yellow, and blue represent distinct replicon types of the plasmids, while all the other replicon types are represented in gray. Source data are provided as a Source Data file.

## National distribution of AmpC β-lactamase genes and plasmids as well as fourth-generation cephalosporin susceptibility

Among the 4,088 strains resistant to 3GCs ((1) in Fig. 1), 1041 harbored AmpC β-lactamase genes, of which 120 strains (11.5%) also harbored ESBL genes (Supplementary Fig. 12a). The 1041 strains possess plasmid-encoded and chromosomal AmpC genes, as 53% of them were identified as either *Enterobacter*, *Citrobacter*, *Morganella morganii*, or *Klebsiella aerogenes*, which are known to harbor chromosomal AmpC genes[21,22]. Among the 2314 ESBL strains resistant to 3GCs ((2) in Fig. 1), 120 (5.2%) also harbored AmpC β-lactamase genes (Supplementary Fig. 12a). These results indicated a substantial fraction of 3GC resistance that was explained not only by ESBL but also by AmpC ((4) in Fig. 1). Kendall's W between carriage of an ESBL or AmpC gene and 3GC resistance was 0.58.

Among the 3159 *E. coli* and 1240 *K. pneumoniae* strains subjected to genome sequencing, two dominant AmpC β-lactamase genes were detected, namely $bla_{CMY-2}$ in *E. coli* and $bla_{DHA-1}$ in *E. coli* and *K. pneumoniae* (Supplementary Fig. 13). Moreover, stratification of STs by the AmpC β-lactamase gene revealed that in *E. coli*, ST131, ST1193, and ST69 accounted for approximately 40% of the strains harboring $bla_{CMY-2}$ or $bla_{DHA-1}$, whereas in *K. pneumoniae*, ST35 and ST23 accounted for approximately 20% of the strains harboring $bla_{DHA-1}$ (Supplementary Figs. 14 and 15).

Approximately 60% of the isolates harboring the $bla_{CMY-2}$ gene carried one of the following six types of plasmids: IncI1-Iα, IncIγ, IncB/

O/K/Z, IncX4, IncFII (pHN7A8), or IncC. In subsequent analyses, we regarded IncI1-Iα and IncIγ collectively as the IncI1 group (Fig. 5), which was the most prevalent type of plasmid harboring $bla_{CMY-2}$ in this study. Approximately 40% of the isolates harboring the $bla_{DHA-1}$ gene carried the IncFII or IncC plasmid. Long-read sequencing enabled us to obtain the complete sequences of five and two types of plasmids harboring $bla_{CMY-2}$ and $bla_{DHA-1}$ (nucleotide sequences are available at https://figshare.com/articles/dataset/Complete_plasmid_sequences_encoding_blaCMY-2_and_blaDHA-1/22084622) (Supplementary Fig. 16). The BLAST top hit of the complete IncI1 group (specifically, IncI1-Iα-type) $bla_{CMY-2}$ plasmid sequence was p22C110-2, a plasmid encoding $bla_{CMY-2}$ that was previously identified in a broiler surveillance in Japan[23], whereas that of the IncB/O/K/Z $bla_{CMY-2}$ plasmid was p4809.66, a plasmid of human urine origin[24].

For the *E. coli* and *K. pneumoniae* strains harboring a single $bla_{CMY-2}$ or $bla_{DHA-1}$ gene (Supplementary Fig. 13), genome alignment with each of the complete plasmid sequences obtained in this study revealed that approximately 60% of the strains harboring a single $bla_{CMY-2}$ gene satisfied the criteria of total sequence length aligned against >90% of the complete plasmid sequences and >90% identity in the aligned sequences, whereas approximately 40% of the strains harboring a single $bla_{DHA-1}$ gene met these criteria (Fig. 5). The geographical distribution of the strains satisfying the criteria stratified by the five regions (south-west, west, middle, east, and north) is presented in Supplementary Fig. 17, in which the proportion of IncC-type (Supplementary Fig. 17b)

was significantly enriched in the east and southwest regions among the strains harboring $bla_{DHA-1}$ ($P < 10^{-10}$, Fisher's exact test). These enriched strains were collected from one and two hospitals in the east and southwest regions, respectively.

Regarding $bla_{CMY-2}$, the four frequent STs among the *E. coli* strains were ST131 (22%), ST405 (14%), ST1193 (12%), and ST69 (8%), in which ST405 and ST69 strains harbored $bla_{CMY-2}$ on chromosomes rather than plasmids (Supplementary Fig. 18). Meanwhile, the Southern blot targeting $bla_{DHA-1}$ showed diverse patterns, suggesting that it is encoded on various plasmids (Supplementary Fig. 19a). Long-read sequencing of six randomly selected strains without IncFII and IncC replicons enabled the construction and comparison of the complete plasmid sequences encoding $bla_{DHA-1}$ (Supplementary Fig. 19b). Results showed diverse plasmids, among which $bla_{DHA-1}$ appeared to be transferred as a mobile element sandwiched between IS26 transposases (Supplementary Fig. 19b).

AmpC β-lactamases degrade penicillin and expanded-spectrum cephalosporins, except for cefepime[25]; hence, susceptibility to cefepime among AmpC β-lactamase-producing Enterobacterales is of interest as a treatment option[26]. Among the 397 3GC-resistant *E. coli* strains carrying the AmpC β-lactamase genes, 26 (7%) also possessed an ESBL gene (Supplementary Fig. 12b). In contrast, among 74 3GC-resistant *K. pneumoniae* strains carrying the AmpC β-lactamase genes, 29 strains (39%) possessed an ESBL gene (Supplementary Fig. 12c), which was significantly higher than that of *E. coli* ($p < 10^{-13}$, chi-square test). Antimicrobial susceptibility profiles of the 3GC-resistant *E. coli* and *K. pneumoniae* strains carrying the AmpC β-lactamase genes are shown in Supplementary Figs. 20 and 21, stratified by $bla_{CMY-2}$ and $bla_{DHA-1}$. As a result, 92% and 91% of the *E. coli* strains carrying $bla_{CMY-2}$ and $bla_{DHA-1}$, respectively, were concordant with the predicted phenotype (susceptible to cefepime), while 67% of the *K. pneumoniae* strains carrying $bla_{DHA-1}$ exhibited concordance with it, likely due to the increased proportion of strains also possessing an ESBL gene. Taken together, these results indicate that cefepime is not a reasonable option for the treatment of AmpC β-lactamase-producing *K. pneumoniae*.

## National distribution of carbapenemase genes, plasmids, and carbapenem susceptibility

The major carbapenemase genes with a frequency of >5% were $bla_{IMP-1}$ (62.2%, out of the total number of carbapenemase genes that perfectly matched known alleles; Supplementary Fig. 22), $bla_{IMP-6}$ (12.4%), other $bla_{IMP}$ variants (6.5%), and $bla_{NDM-5}$ (7.4%; Supplementary Fig. 23). In *E. coli*, $bla_{IMP}$ and $bla_{NDM}$ exhibited similar prevalence, whereas $bla_{IMP}$ was dominant in other species. The breakdown of STs among *E. coli*, *K. pneumoniae*, or *Enterobacter* spp. strains, stratified by carbapenemase genes, is shown in Supplementary Dataset 3, including diverse STs and suggesting transfer of plasmids encoding the genes among the STs.

The relationship between the carriage of a carbapenemase gene and meropenem MICs for all carbapenemase genes detected is summarized as a Sankey plot in Fig. 6, revealing that all strains carrying $bla_{IMP-6}$ had MIC values ≥ 2 μg/mL (intermediate or resistant to meropenem), whereas 13.3% of strains carrying $bla_{IMP-1}$ showed MIC values ≤ 1 μg/mL (susceptible). Overall, Kendall's W between the carriage of a carbapenemase gene and meropenem-resistant/intermediate/susceptible phenotype was 0.85 among the 4088 strains resistant to 3GCs. Two of the strains carrying $bla_{IMP-6}$ exhibited MIC values ≤ 1 μg/mL, according to initial antimicrobial susceptibility testing in the hospital laboratories where they were isolated. Thus, considerable variability in MIC values was observed, demonstrating the importance of standardized MIC measurement protocols in AMR surveillance.

All strains carrying $bla_{NDM}$ (orange in Fig. 6, enriched in *E. coli*) were resistant to meropenem, whereas those carrying $bla_{OXA-181}$, $bla_{OXA-48}$, or $bla_{VIM-1}$ were susceptible. Another Sankey plot for the MIC of

imipenem is presented in Supplementary Fig. 24, confirming that most strains carrying $bla_{IMP-6}$ showed imipenem MIC values ≤ 0.5 μg/mL (susceptible, as previous reported[27]; Supplementary Fig. 24). Analysis of the distribution of carbapenem MIC values (imipenem, meropenem, and doripenem) for strains harboring $bla_{IMP-1}$ and $bla_{IMP-6}$ revealed that no strain harboring $bla_{IMP-6}$ exhibited meropenem MIC ≤ 1 μg/mL, whereas 13.3% of strains harboring $bla_{IMP-1}$ had meropenem MIC values of 1 or ≤ 0.5 μg/mL (Supplementary Fig. 25). Additionally, susceptibility to AMK was highly preserved in both strains with $bla_{IMP-1}$ and $bla_{IMP-6}$ (>95% susceptibility; Supplementary Fig. 26). Subsequently, the relationship between specimen sources, STs, and number of strains harboring $bla_{IMP-1}$ or $bla_{IMP-6}$ was visualized for *E. coli* and *K. pneumoniae* (Supplementary Figs. 27 and 28). The *E. coli* strains harboring $bla_{IMP-1}$ or $bla_{IMP-6}$ were primarily ST131 and isolated from either urine (mostly harboring $bla_{IMP-1}$), blood (all harboring $bla_{IMP-6}$), or respiratory specimens (all harboring $bla_{IMP-1}$). In contrast, among the *K. pneumoniae* strains, the two major STs were ST12 and ST15, and the two major specimen sources were sputum and stool.

Systematic long-read sequencing of isolates harboring $bla_{IMP-1}$, $bla_{IMP-6}$, or $bla_{NDM-5}$ enabled the reconstruction of 183 complete plasmid sequences encoding the three major carbapenemase genes (Supplementary Dataset 4) (nucleotide sequences are available at https://figshare.com/articles/dataset/Complete_plasmid_sequences_encoding_blaIMP-1_blaIMP-6_and_blaNDM-5/22084769). The dataset of complete plasmids and other plasmids whose sizes and replicons were determined was used to elucidate their geographical distribution stratified by replicon types and species carrying the plasmids (Fig. 7). IncN plasmids encoding $bla_{IMP-6}$ were clustered in the western region of Japan, as previously reported[27,28], with 62% carried by *Klebsiella* spp., comprising 11 STs. These results suggest the transfer of plasmids encoding $bla_{IMP-6}$ in the western region among STs within *Klebsiella* spp. as well as among different species. In addition, we discovered that IncHI2 plasmids encoded $bla_{IMP-6}$. In particular, IncHI2 plasmids were carried by three strains isolated in the eastern region, all of which also encoded *mcr-9* (mobilized colistin resistance) genes, although the isolates carrying these plasmids were susceptible to colistin. Three other plasmid types (FIA-FIB-N, FIA-FII-N, and FIB-N) harboring $bla_{IMP-6}$ were also identified as multi-replicon plasmids containing the IncN replicon. These plasmids harbored at least 90% of the sequence of a typical $bla_{IMP-6}$-harboring plasmid, pKPI-6[27], suggesting that they have emerged as a result of plasmid fusion between the IncN-type and other plasmids coexisting in the isolates. In contrast, plasmids encoding $bla_{IMP-1}$ had a higher diversity of replicons than those encoding $bla_{IMP-6}$. Notably, IncHI2 plasmids encoding $bla_{IMP-1}$ were primarily carried by *Enterobacter* spp. and were widely distributed in eastern Japan (Fig. 7). IncHI2 plasmids were isolated in 26 different hospitals; however, no more than five were collected from any one hospital. In contrast, IncN, M1, and M2 plasmids encoding $bla_{IMP-1}$ were detected in multiple species in a single hospital (Fig. 7), suggesting cross-species transfer of the plasmids in a hospital. IncFIA, IncFIA-R, IncFIB/FII, and IncFII (pECLA) plasmids encoding $bla_{IMP-1}$ were enriched in the southwest region (top left in Fig. 7). Meanwhile, plasmids encoding $bla_{NDM-5}$, consisting of five replicon types, were sporadically distributed across the country.

Complete sequences of the plasmids carrying major replicons with the characteristics described above (IncN, HI2, M1, and M2) were used to detect all AMR genes across the plasmid sequences. These data were combined with information on plasmid size, antimicrobial susceptibility profiles, host bacterial species, and geographical regions where they were isolated (Supplementary Fig. 29). In the plasmids harboring $bla_{IMP-1}$, a significant association was observed with the presence or absence of the following AMR gene combinations ($P_{FDR} < 10^{-7}$, Fisher's exact test) (Supplementary Fig. 29): (1) *aac(6')-set_A*, $bla_{CTX-M-2}$, and *tet*(A); (2) *aac(6')-Ilc*, *qnrB6*, and *tet*(B); (3) *aac(6')-Ib-G* and *aadA24*. Similarly, in the plasmids harboring $bla_{IMP-6}$, a

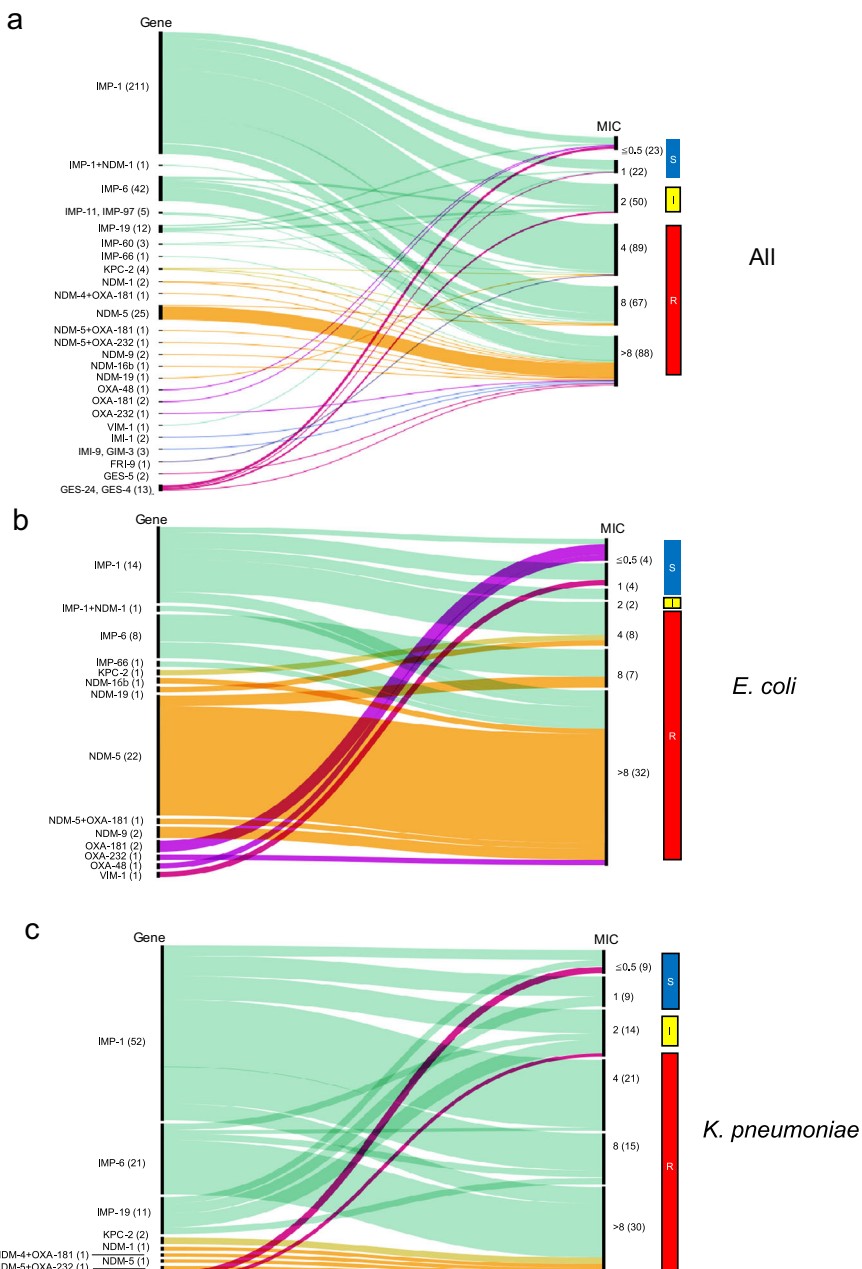

**Fig. 6 | Sankey plot summarizing the relationship between carriage of a carbapenemase gene and meropenem MIC. a** All species with a carbapenemase gene; **b** *E. coli*; **c** *K. pneumoniae*. The numbers in the parentheses indicate the number of strains carrying the gene (at the left) and showing the MIC (at the right). Source data are provided as a Source Data file.

significant association was detected with the following AMR gene combinations ($P_{FDR} < 0.05$, Fisher's exact test): (1) $bla_{CTX-M-2}$ and *tet*(A) (Supplementary Fig. 30); (2) $bla_{CTX-M-9}$, *mcr*, and *qnrA1*. Each combination corresponds to a specific Inc type (IncN, IncHI2, and IncM2 in Supplementary Figs. 29 and 30) and can confer resistance to multiple classes of antimicrobials that are potentially caused by a single transfer of this plasmid.

Relationships between replicon types and plasmid sizes are plotted in Supplementary Fig. 31, indicating that IncHI2 plasmids harboring $bla_{IMP-1}$ were generally larger than the others.

We also detected the rare carbapenemase genes $bla_{FRI-9}$ ($N = 1$), $bla_{IMI-1}$ ($N = 2$), and $bla_{IMI-9}$ ($N = 2$). Long-read sequencing of strains harboring these genes revealed that $bla_{FRI-9}$ was located in an IncFIA plasmid that exhibited synteny with another plasmid obtained from a

clinical strain in Osaka, Japan (Supplementary Fig. 32a), although it lost the genes responsible for conjugation. Moreover, four $bla_{IMI}$ genes were located in their chromosomes and were carried by one of three types of integrative mobile genetic elements exploiting the Xer-mediated recombination mechanism (IMEX): EcloIMEX-2, -3, or -5[29] (Supplementary Fig. 32b).

## Discussion

We aimed to conduct a national genomic surveillance of Enter-obacterales resistant to 3GCs and those satisfying the epidemiological cutoff for screening carbapenemase, integrating standardized quantitative antimicrobial susceptibility testing. Our wide-range re-measurement of MIC was conducted using the same panels in the same location, which generated the largest ($N = 4195$) quantitative dataset of

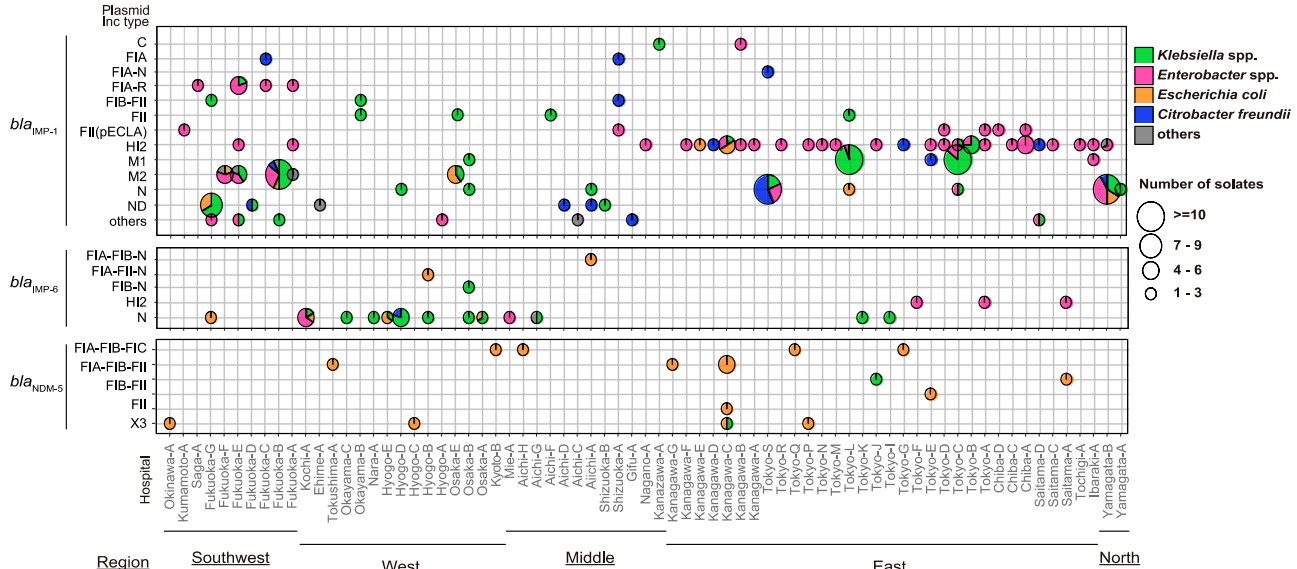

**Fig. 7 | Geographical distribution of *bla*ₗₘₚ₋₁, *bla*ₗₘₚ₋₆, and *bla*ₙDₘ genes stratified by replicon type and species carrying the plasmids.** Vertical lines correspond to hospitals; horizontal lines correspond to replicon plasmids. The size of the circles reflects the number of strains (*N* = 272, carrying complete plasmids and other plasmids whose sizes and replicons were determined) carrying the replicon type and isolated in each hospital. Source data are provided as a Source Data file.

MIC values. The associated dataset and corresponding genome sequence data have been made publicly available and will prove useful in future studies, for example, when constructing machine learning models for predicting antimicrobial resistance from genetic data[30,31]. In our study, combined analysis of the phenotypic and genomic datasets provided a complete classification of 3GC resistance mechanisms (Fig. 1), including those, other than ESBL, that are not commonly considered in clinical laboratories. Notably, among strains resistant to 3GCs, 43.5% (1,774/4,088) lacked ESBL genes, which was overestimated in the present study because of the selection of strains for genome sequencing. We selected all isolates that did not grow in CHROMagar ESBL culture to explore mechanisms, other than ESBL, underlying resistance to 3GCs.

The combined analysis for carbapenems revealed the relationship between carriage of a carbapenemase gene and the meropenem (Fig. 6) or imipenem (Supplementary Fig. 24) MICs for all detected carbapenemase genes. In Japan, carbapenem resistance of Enterobacterales is defined as a meropenem MIC ≥ 2 μg/mL (or imipenem MIC ≥ 2 μg/mL and cefmetazole MIC ≥ 64 μg/mL), which corresponds to 86.7% (88.5% if the criterion using imipenem and cefmetazole was also applied) of the 339 isolates harboring carbapenemase genes detected in our study; i.e., 13.3% of the isolates harboring carbapenemase genes could not be detected using the designated criteria, thus emphasizing the importance of genomic surveillance, including strains with lower MICs, to monitor the prevalence and spread of carbapenemase genes.

Extensive genome sequencing identified 339 isolates harboring carbapenemase genes, of which 82.3% (279 isolates) were collected by the hospitals participating in this surveillance to satisfy the epidemiological cutoff (meropenem MIC ≥ 0.25 μg/mL), confirming the high sensitivity of the epidemiological cutoff for screening carbapenemase genes. Among the remaining 60 isolates harboring carbapenemase genes that did not satisfy the epidemiological cutoff at the hospitals, 49 isolates had a meropenem MIC ≥ 1 μg/mL after the re-measurement of MIC at the National Institute of Infectious Diseases. The remaining 11 isolates had a meropenem MIC ≤ 0.5 μg/mL (the lowest limit of the panel used in our study), and another panel was required to determine the lower MIC.

The combined analysis also generated charts summarizing the degree of susceptibility to each antimicrobial for *E. coli* and

*K. pneumoniae* strains harboring ESBL genes (Supplementary Fig. 9), *E. coli* strains harboring ESBL genes stratified by major *bla*CTX-M genes (Supplementary Fig. 10), *E. coli* and *K. pneumoniae* strains harboring AmpC β-lactamase genes stratified by the presence of *bla*CMY-2 and *bla*DHA-1 (Supplementary Figs. 20 and 21), and Enterobacterales strains harboring the IMP carbapenemase genes stratified by the presence of *bla*IMP-1 and *bla*IMP-6 (Supplementary Fig. 26). These charts could help guide decisions regarding optimal treatment regimens for ESBL- or AmpC-producing or carbapenem-resistant Enterobacterales infections in Japan.

Our systematic long-read sequencing analysis revealed the diversity of plasmid replicon types and the specific relationships between replicon types, host bacterial species, and geography and plasmids encoding *bla*IMP-1, *bla*IMP-6, and *bla*NDM-5 (Fig. 7). In addition, for each major ESBL and AmpC β-lactamase gene, we prepared datasets for reference plasmids carrying the gene and explored the extent by which the presence of the gene could be explained based on a comparison of complete plasmid sequences reconstructed from long-read sequencing data with the genome sequence of each isolate (Figs. 3, 5, Supplementary Fig. 7). Except for *bla*CTX-M-8, which is rare among the *bla*CTX-M genes (Fig. 2), the extent explained by the reference plasmids did not account for more than 60% (Figs. 3 and 5), suggesting a high diversity in plasmids or genetic elements that can carry the major ESBL and AmpC β-lactamase genes, including those not yet included in our reference plasmids. Repeating the process when additional complete plasmid sequences are obtained in future studies will continue to improve the dataset for reference plasmids that can carry the genes, which will help enhance our understanding of how genes are transferred via plasmids.

Our genomic surveillance (the Japan Antimicrobial Resistant Bacterial Surveillance, JARBS) was designed as a research project to be linked to the established sustainable national surveillance program, "Japan nosocomial infections surveillance (JANIS)"[18]. JANIS is funded by the Ministry of Health, Labour, and Welfare (MHLW) and managed by the National Institute of Infectious Diseases. It collects all routine bacterial culturing and antimicrobial susceptibility testing results, including both culture-positive and culture-negative data of all sample types. Hospitals participating in JARBS were recruited from those participating in JANIS, and each strain collected in the hospitals had

specimen IDs linked to the JANIS database. The genomic analysis of strains collected in JARBS is integrated at the National Institute of Infectious Diseases, thereby complementing the comprehensive phenotypic surveillance performed by JANIS, and enabling data linkage between the two databases. To ensure the sustainability of the integrated genomic analysis over time, we have made continuous efforts to secure financial support from the MHLW as stated in the new national action plan on AMR starting in 2023. Moreover, we provide feedback reports to each participating hospital, informing them about the presence or absence of major antimicrobial-resistant genes in their respective strains. This serves as an incentive for hospitals to maintain their participation in genomic surveillance. Furthermore, we initiated the second phase of JARBS in March 2023.

The comprehensive data collection scheme enabled JANIS to determine the surveillance denominator, which could be applied to calculate the prevalence of AMR, as requested by the WHO Global Antimicrobial Resistance and Use Surveillance System (GLASS). Although the denominator of surveillance is generally missing in genomic surveillance, linking data to the JANIS database will enable estimation of the prevalence of AMR determinants detected in genome sequence data. Furthermore, the data linkage between our genomic and phenotypic surveillance can be extended to epidemiological and clinical data via anonymized patient IDs. Such an extension of our national AMR surveillance could enable combined analyses of the clinical and social burdens of AMR determinants, which could direct public health actions to control AMR.

In summary, after combining the genome data and standardized quantitative antimicrobial susceptibility testing results for 4,195 Enterobacterales isolates collected across Japan, we provided a comprehensive classification of 3GC resistance mechanisms and illustrated the relationship between the carriage of carbapenemase genes and carbapenem MICs, highlighting genes that cannot be detected using routine susceptibility testing. We also created datasets for reference plasmids carrying major ESBL or AmpC β-lactamase genes and revealed the extent to which the reference plasmids can explain the national distribution of ESBL and AmpC β-lactamase genes. Our systematic long-read sequencing further constructed 183 complete plasmids encoding the three major carbapenemase genes and elucidated detailed relationships between specific plasmid Inc types and the frequency of carbapenemase and other AMR genes, geographical distribution, host species, and potential plasmid transfer events that could cause resistance to multiple classes of antimicrobials. In summary, our study provides a blueprint for national genomic surveillance that integrates standardized quantitative antimicrobial susceptibility testing, while also characterizing the determinants of 3GC and carbapenem resistance, all of which can be linked to other data sources at the national level.

## Methods

### Isolates and DNA sequencing
*E. coli* and *K. pneumoniae* isolates resistant to 3GCs (either cefotaxime (CTX), ceftazidime (CAZ), ceftriaxone (CTRX), or cefpodoxime (CPDX)) and Enterobacterales isolates with reduced susceptibility to meropenem (MEPM) showing MIC ≥ 0.25 μg/mL (i.e., the epidemiological cutoff defined in EUCAST for screening of carbapenemase) were collected in 2019 and 2020 from 175 hospitals based on antimicrobial susceptibility testing results in each hospital across Japan (covering 45 out of 47 prefectures) through the Japan Antimicrobial Resistant Bacterial Surveillance (JARBS)[18]. The National Institute of Infectious Diseases conducted the following procedures from initial screening to systematic short- and long-read sequencing for the collected isolates. All collected isolates ($N = 23,295$) were initially screened by performing the multiplex PCR assay using PCR with the QIAGEN Multiplex PCR Plus Kit (QIAGEN, Germany) and specific primers for the ESBL genes ($bla_{CTX-M}$ group 1, 2, 8, 9, $bla_{TEM}$, and $bla_{SHV}$)[32] and carbapenemase

genes ($bla_{KPC}$, $bla_{IMP}$, $bla_{NDM}$, $bla_{VIM}$, $bla_{OXA-48}$-like, and $bla_{GES-5}$-like)[33]. In addition, we used another set of primers for the confirmation of carbapenemase genes in strains that showed questionable results[34–37]. The boiling lysis method was used for preparing the template for PCR. Colonies were resuspended in 0.1 mL of molecular biology-grade water, boiled at 100 °C for 10 min, cooled on ice, and centrifuged at $7340 \times g$ for 2 min; 50 μL of the supernatant was stored at −30 °C. The PCR products were visualized using a Microchip Electrophoresis System (MultiNA; Shimadzu, Kyoto, Japan) with a DNA-1000 Reagent Kit (Shimadzu), following the manufacturer's recommendations. Isolates grown in CHROMagar ESBL culture were cultured in Muller-Hinton broth overnight. Bacteria were lysed with 0.5 mg/mL lysozyme and 2% SDS. Genomic DNA was purified from the lysate using AMPure XP beads (Beckman Coulter, USA). DNA libraries were prepared using an Enzymatics 5× WGS fragmentation mix and WGS ligase reagents (Qiagen, Hilden, Germany). Paired-end sequencing (2×150 bp) was performed on the Illumina HiSeq X FIVE platform (Macrogen Japan Corporation, Tokyo, Japan) for 5,143 isolates that were selected based on the PCR results to include isolates that 1) were carbapenemase gene-positive, 2) were carbapenemase gene-negative with reduced susceptibility to carbapenems according to antimicrobial susceptibility testing results, 3) were representative in terms of ESBL gene carriage patterns; 4) did not grow in CHROMagar ESBL culture, suggesting other mechanisms underlying resistance to 3GCs.

Isolates positive for carbapenemase genes were analyzed using S1-PFGE and Southern blot analysis targeting each carbapenemase gene[38]. PFGE plugs treated with S1 nuclease (Takara Bio, Shiga, Japan) were subjected to PFGE using the CHEF Mapper XA system (Bio-Rad, Hercules, CA, USA). The separated DNA was transferred to a nylon membrane, and those containing the carbapenemase gene were detected using digoxigenin-labeled DNA probes (Roche Diagnostics, Basel, Switzerland) specific for $bla_{IMP}$, $bla_{NDM}$, or $bla_{IMI}$. Long-read sequencing analysis using the GridION system (Oxford Nanopore Technologies, UK) was also conducted for the isolates positive for carbapenemase genes. We selected at least one isolate among those harboring the same size plasmid encoding the same carbapenemase gene in a hospital for long-read sequencing. For the carbapenemase gene-carrying plasmids that were not subjected to long-read sequencing analysis, the plasmid type was determined to be the same as that of a completed plasmid originating from the same hospital if these plasmids had the same size and the replicon carried by the completed plasmid was detected in the short-read sequencing data of the isolate carrying the compared plasmid. We also conducted long-read sequencing for representative strains harboring the major AmpC β-lactamase genes $bla_{CMY-2}$ and $bla_{DHA-1}$. Genomic DNA was prepared using a Genomic tip (Qiagen, Germany) or Monarch HMW DNA Extraction Kit for Tissue (New England BioLabs, USA). Libraries were prepared using the Rapid Barcoding Kit, and sequencing was carried out with the R9.4.1 flow cell. Base calling was performed using Guppy (v4.0.11-v5.0.12), and hybrid assembly was conducted using Unicycler (v0.4.8)[39]. When the assembly containing the carbapenemase gene was not completed using Unicycler, Flye (v2.8.2 or v2.9)[40] was used to assemble GridION sequence reads, and assembled contigs were polished by Illumina reads using Pilon version 1.24 (5).

### Antimicrobial susceptibility testing
Of the 5143 newly sequenced strains, the MICs of 4195 strains were determined at the National Institute of Infectious Diseases using the broth microdilution method implemented in MicroScan WalkAway (Beckman Coulter Inc.) and the NEG MIC 3.31E and NEG MIC NF 1J panels (Beckman Coulter Inc.). Not all of the newly sequenced strains were subjected to antimicrobial susceptibility testing because a sufficient number of results were obtained for certain types of ESBL isolates (e.g., carrying the CTX-M-9 group gene). Antimicrobial susceptibility testing was designed to measure major antimicrobials using five

dilutions to cover a therapeutically achievable range[41] (Supplementary Dataset 5): ampicillin/sulbactam, piperacillin, piperacillin/tazobactam, cefmetazole, cefoxitin, cefotetan, cefpodoxime, cefotaxime, cefotaxime/clavulanate, ceftriaxone, ceftazidime, ceftazidime/clavulanate, cefoperazone/sulbactam, aztreonam, cefozopran, cefepime, imipenem, meropenem, doripenem hydrate, gentamicin, tobramycin, amikacin, levofloxacin, ciprofloxacin, minocycline, tigecycline, fosfomycin, sulfamethoxazole/trimethoprim, colistin, and chloramphenicol. The abbreviations for the antimicrobials are shown in Supplementary Dataset 5. The NEG MIC 3.31E panel requires manual inspection to determine MIC values; for antibacterial drugs that had results from both panels, the MIC values measured using the NEG MIC NF 1J panel were tabulated.

To define susceptible/resistant phenotypes, MIC cutoffs were used according to the Clinical and Laboratory Standards Institute[42,43] 2021. The cutoff changed only for piperacillin/tazobactam from CLSI 2021 to 2022.

For *E. coli* strains that were resistant to 3GCs but did not harbor an acquired gene responsible for resistance to 3GCs, we examined expression of the chromosomal AmpC gene that is normally constitutively expressed at low levels[9,44] using the CPDX disk diffusion method. In brief, a CPDX disk was placed on the surface of the agar containing the cultured strains. Next, 10 μL of 3-aminophenyl boronic acid (APB) and cloxacillin (MPIPC) were added. A more than 5-mm increase in the zone diameter on plates containing APB and MPIPC indicated that a high level of AmpC enzyme was produced.

### Bioinformatic analyses

Low-quality reads were removed using fastp[45] followed by genome assembly using Shovill[46]. The number of contigs and N50 of each isolate were then determined (Supplementary Dataset 6). The taxonomy of each genome was checked using DFAST_QC[47] and MLST typing was conducted using mlst[48] (https://github.com/tseemann/mlst) or an in-house Perl script to detect variants[49]. For detection of genetic AMR determinants, we combined the Bacterial Antimicrobial Resistance Reference Gene Database in NCBI[13] with the ResFinder database[50]. Nucleotide sequences in the latter were added to the former if a BLASTn match showed <60% locus length or <100% sequence identity with any nucleotide sequence in the former. We then conducted a BLASTn search of each genome against the combined database as well as a DIAMOND[13] search of the translated version of the combined database. Matches of >80% identity and >60% locus length were set as the criteria for positive detection of the gene and a perfect match of amino acid sequence to that of a specific gene type. Automatic classification of carbapenemase-type genes (e.g., $bla_{OXA-48}$, $bla_{OXA-181}$, $bla_{OXA-232}$, $bla_{GES-5}$, $bla_{GES-16}$, and $bla_{GES-24}$) and others among $bla_{OXA}$ and $bla_{GES}$, as well as that of ESBL type genes (e.g., $bla_{SHV-17}$, $bla_{SHV-2}$, and $bla_{TEM-19}$) and others among $bla_{TEM}$ and $bla_{SHV}$, were conducted according to BLDB[51] (Supplementary Dataset 1). ESBL was defined as Ambler class A, phenotype 2be, and functional information ESBL in BLDB. CPE were defined as harboring $bla_{IMP}$, $bla_{KPC}$, $bla_{NDM}$, $bla_{IMP}$, $bla_{VIM}$, $bla_{POM}$, $bla_{TMB}$, $bla_{DIM}$, $bla_{FIM}$, $bla_{HMB}$, $bla_{SMB}$, $bla_{IMI}$, $bla_{FRI}$, $bla_{GIM}$, and carbapenemase-types $bla_{OXA}$ and $bla_{GES}$. A BLASTn search was also conducted against the nucleotide sequences of the genes encoding outer membrane porins *ompA*, *ompF*, *ompC*, and *phoE* in *K. pneumoniae*; genes *ompC* and *ompF* in *E. coli*; and genes encoding Omp35 and Omp36 in *K. aerogenes*. Additionally, mutations in the promoter region of the chromosomal AmpC ($bla_{EC}$) gene were identified and evaluated using AMRFinderPlus[13].

Plasmid replicons were detected using PlasmidFinder[52] with default parameters, or abricate, or MOB-typer[53]. For the carbapenemase gene-encoding plasmids that were not subjected to long-read sequencing analysis, the plasmid type was determined to be the same as that of a completed plasmid originating from one hospital if the size of the plasmids was the same, and the replicon carried by the

completed plasmid was detected in the short-read sequencing data of the isolate harboring a compared plasmid.

Genome alignment between an isolate and each of the 25 complete reference plasmid sequences encoding $bla_{CTX-M-2}$, $bla_{CTX-M-14}$, $bla_{CTX-M-27}$, $bla_{CTX-M-8}$, $bla_{CTX-M-3}$, $bla_{CTX-M-15}$, or $bla_{CTX-M-55}$ obtained in an ESBL surveillance in Hiroshima, Japan (Data availability) was conducted using progressiveMauve[54], from which the percentage of aligned bases and nucleotide sequence identity in the alignment were calculated. Similarly, we conducted genome alignment between an isolate and complete plasmid sequences that encode $bla_{CMY-2}$ and $bla_{DHA-1}$, respectively, and were newly decoded in this study. This genome alignment approach using progressiveMauve does not exclude the possibility of contigs carrying an AMR gene breaking at points where repeat regions collapse due to mobile genetic elements.

Visual genome comparison among complete plasmid sequences was conducted using EasyFig[55]. A network plot of specific antimicrobial resistance genes and antimicrobial susceptibilities was illustrated using Gephi[56] and output files of Staramr[57]. The network layout was obtained using ForceAtlas2[56], a force-directed continuous algorithm under scaling set to 1 and null edge weight influence. Phylogenetic trees were constructed using FastTree 2[58] and metadata were visualized using Microreact[59].

Kendall's W was calculated using R statistical software 4.1.2 and the KendallW function in DescTools package. The Sankey plots were illustrated using Rawgraph (https://app.rawgraphs.io).

### Statistics & reproducibility

No statistical method was used to predetermine sample size. No data were excluded from the analyses. No data were excluded from the analyses of 5143 newly sequence strains in which the MICs of 4195 strains were determined. This study is reproducible from the publicly available genome sequence data and the metadata for each isolate summarized in Supplementary Dataset 6.

### Ethics

This study was approved by the IRB of the National Institute of Infectious Diseases (approval number: 1251). Approval for the use of JANIS data linked to the strains collected in the hospitals was granted by the Ministry of Health, Labor and Welfare of Japan (approval number: 1553).

### Reporting summary

Further information on research design is available in the Nature Portfolio Reporting Summary linked to this article.

## Data availability

The metadata for each isolate, including MLST, MIC, and genetic polymorphisms, are summarized in Supplementary Dataset 6. Phylogenetic trees together with metadata visualized using Microreact are available at https://microreact.org/project/piQLyJmufXmM7gwCYzw6eN-microreactecoli3158up2022-04-25 for *E. coli* (Supplementary Fig. 4) and https://microreact.org/project/2RRmtHGa74444NZHPivC4h-microreactkp1240up2022-04-26 for *K. pneumoniae* (Supplementary Fig. 5). BLDB was used for classification of carbapenemase-type genes and others among $bla_{OXA}$ and $bla_{GES}$, as well as that of ESBL type genes and others among $bla_{TEM}$ and $bla_{SHV}$. The 25 complete plasmid sequences encoding $bla_{CTX-M-2}$, $bla_{CTX-M-14}$, $bla_{CTX-M-27}$, $bla_{CTX-M-8}$, $bla_{CTX-M-3}$, $bla_{CTX-M-15}$, or $bla_{CTX-M-55}$ were deposited in DDBJ under accession numbers DRA014676 to DRA014704 (BioSample accession numbers SAMD00521005 to SAMD00521033, respectively) (https://ddbj.nig.ac.jp/resource/bioproject/PRJDB10032). The raw short- and long-read data of the newly sequenced strains were deposited at DDBJ under the BioProject accession number PRJDB10842. Source data are provided with this paper.

## Code availability

The custom code used in this study is available at https://github.com/bioprojects/JARBS-GNR[60].

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

## Acknowledgements

Computational calculations were performed at the Human Genome Center of the Institute of Medical Science (University of Tokyo) and the National Institute of Genetics. We are grateful to all the hospitals participating in JARBS (https://www.niid.go.jp/niid/images/amrc/JARBS1_participaring_175hospitals_v2.xlsx). We are also grateful to the following staff members for their technical contributions to this project: Eiko Anzai, Emi Fujimura, Takahisa Ishizuka, Koichi Shimakawa, Rumi Oki, Yoshie Taki, Satoyo Wakai, Sadao Aoki, Mayumi Sasada, Mikihisa Okuda, Akira Moriya, Sayaka Uchino, Wataru Hayashi, Mikako Nakazawa, Noriko Sakamoto, Elahi Shaheem, Chika Arai, Yuko Kazumi, Akimi Suzuki, and Fumiko Hamamoto. This work was supported by the Research Program on Emerging and Re-emerging Infectious Diseases of the Japan Agency for Medical Research and Development (AMED) under Grant Number 23fk0108604 (to M.Sugai).

## Author contributions

M.Sugai conceptualized this study. M.Sugai, S.Kayama, J.H., and S.Kawakami designed this study. S.Kayama, S.Kawakami, A.H., M.Suzuki, and T.K. participated in the PCR screening of all collected strains. S.Kayama, S.Kutsuno, H.K. and K.K. were involved in species identification. S.Kayama, Y.S., H.Z., N.K., L.Y., M.Suzuki, and J.H. contributed to genome sequencing. S.Kayama, A.H., N.K., and T.K. contributed to antimicrobial susceptibility testing. S.Kayama, Y.S., S.Kawakami, K.K. and K.Y. analyzed the data. K.Y. and N.K. contributed to the information processing and database construction. K.Y. was the major contributor to the writing of the manuscript. All authors have read and approved the final manuscript.

## Competing interests

The authors declare no competing interests.
