## [Peer Review File · Nature Communications]

REVIEWER COMMENTS

Reviewer #1 (Remarks to the Author):

The manuscript describes the use of whole-genome sequencing (WGS) to add value to Japan's national AMR resistance surveillance system (JANIS) in a retrospective study. The study shows how genomics surveillance can clarify antimicrobial resistance mechanisms and provide a complete classification of 3rd generation cephalosporins. It also uses systematic long read sequencing to characterize representative plasmids outside of previous reference country plasmids and allowed their geographical distribution and highlight potential plasmid transfer events.

The manuscript suggests the generation of a blueprint for the integration of genomic epidemiology into a laboratory-based surveillance system in a retrospective study at the National Reference Laboratory (NRL) level, but the description of how this is sustainable through time is not clearly stated, neither how the genomic analysis is integrated at the NRL. Considering the title, this description should be further developed.

This manuscript adds value in comparison to other publications describing the use of WGS in AMR surveillance (1, 2), particularly with a dataset of great quality considering that it is well phenotypically characterized and confirmed at the NRL with further short reads and long reads study.

The methodology is perfectly described and all software used are publicly available which is essential for reproducibility.

In line 255 where Figure 6a is mentioned as a Sankey plot reveals that most strains carrying blaIMP-6 had IMC values ≥ 2 ug/ml, this is not clear, whereas it is stated in Supplementary figure 20.

In conclusion, the manuscript is clear and accessible, with good and complete references and reproducible and access data. The abstract perfectly summarizes the content of the work. It would add original value to this work the expanded description of integration of WGS in a blueprint of a very well established AMR national surveillance system with systematic short and long read data and provide a dataset linked to other data sources for further studies.

1- Koser CU, Ellington MJ, Peacock SJ. Whole-genome sequencing to control antimicrobial resistance. *Trends Genet.* 2014;30(9):401-7.

2- Argimon, S. et al. Integrating whole-genome sequencing within the National

Antimicrobial Resistance Surveillance Program in the Philippines. *Nat Commun* 11, 586 2719 (2020). <https://doi.org/10.1038/s41467-020-16322-5>

Reviewer #2 (Remarks to the Author):

Kayama et al. provide an extensive molecular epidemiological study of third generation cephalosporin (3GC) Enterobacterales with reduced meropenem susceptibility in Japan. The authors importantly were able to confirm narrow-spectrum beta-lactam, beta-lactam/beta-lactamase inhibitor (BL/BLI), 3GC and carbapenem AST phenotypes with 4195 sequenced Enterobacterales species providing a robust analyses of AMR gene carriage and associations with particular clinically relevant AST phenotypes as well as plasmid vectors.

These analyses are critical to understand genotype/phenotype correlations across multiple geographical regions. Crucially, highlighting concordance and importantly discordance with susceptibility tested versus AMR gene detection is important to inform potential molecular diagnostic measures. This is illustrated by the fact that 15% of carbapenemase positive isolates had carbapenem susceptible profiles (Fig 6 and Fig. S20). While these findings are well documented throughout the manuscript, I believe that a concordance measure of AMR gene vs. AST phenotype (e.g., Kendall's W) would be a great quantification and presentation of the results that are shown primarily through figures that are often difficult to assess on their own. I think this could simply be focused on some of the important antibiotic classes, such as carbapenemases as well as BL/BLIs to name a few. While perhaps some figure legends could also include additional detail, most of the figures provide nice visual aids to make sense of particular AMR gene relationships. Additionally, the methods are well described with great supporting metadata as they mention could be of great use to persons using publicly available data to build prediction models. Below are some further comments that could further aid in clarification/interpretation as well as minor editing suggestions.

Major Comments:

Line 104 – 148: There are inherent limitations to inferring ESBL encoding gene genomic context with reference-based alignments that should be considered. While the co-occurrence of ESBL presence with a certain ID/COV% of a select few reference plasmids does suggest the potential of each respective plasmid as a vector for CTX-M carriage, these interpretations should only be stated as associations or 'likely carriage' unless there is further evidence of co-carriage within each plasmid (i.e., additional long-read sequencing or more extensive S1-PFGE + SB) especially at this large of a scale. If authors want to use a 90% cutoff threshold to determine 'likely carriage', at the least, inferring genomic context with short-read alignments should be mentioned as a limitation. This has been addressed in part in line 383 – 390, but should mention that this threshold cutoff does not rule out potential for other ESBL encoding gene genomic contexts.

Line 171-179: While the network graph in Fig. S10 does a decent job of visually showing nodal sharing between blaTEM-1b and blaCTX-M-14/blaCTX-M-15 in contrast to blaCTX-M-27, it may be useful to

present blaTEM-1b presence/absence data in a complimentary table (with co-occurrence of each respective blaCTX-M variant) to demonstrate the relationship with SAM susceptibility.

Line 237 – 243: The difference in FEP susceptibilities doesn't indicate that *K. pneumoniae* (Kp) co-carried an ESBL gene with blaDHA-1. I would report the number of isolates that carry both ESBL + blaDHA-1 29/74 (39%) in Kp as reported in Fig. S11 and how many are concordant with predicted phenotypes as shown in Fig. S18.

Minor Comments:

Line 90: May be helpful to include frequency counts in supplemental table 1 for each SHV, TEM, GES variant, otherwise not sure how helpful the inclusion of this table is to the manuscript.

Line 97 – 99: Was there any assessment of likely AmpC derepression through assessment of qRT-PCR and/or ampC promoter mutations? The assessment of ampC promoter mutations using AMRFinder plus could potentially help clarify the genetic mechanisms.

Line 165; 168: Should spell out levofloxacin and ciprofloxacin

Line 187: Would clarify these are plasmid encoded AmpC genes

Line 201: Should be IncFII

Line 224: The detection of plasmid replicon types enriched in particular hospitals does not suggest an outbreak. Sequence similarity between isolates harboring such plasmids would need to be included with additional temporal/spatial epidemiological data.

Line 248: Would clarify how percentages are being calculated, i.e., the percent of total carbapenemase

Line 270: I believe that "AMK" is supposed to be "Augmentin – amoxicillin-clavulanate". However, it is commonly used as an abbreviation for amikacin, an aminoglycoside antibiotic, so would be careful with nomenclature. In the past, I have used the abbreviation "AMC" to avoid this confusion.

Line 427: Did non-E. coli/K. Pneumoniae Enterobacterales species have different selection criteria, i.e., 3GC AST was not performed, rather reduced MEM susceptibility was inclusion criteria? If so, is there a reason why or is this due to how isolates are surveilled through JARBS?

Line 532: To elaborate on previous point, progressiveMauve alignments of long- to short-read assemblies will still not definitively determine for a proportion of assemblies whether a colinear block that carries an ESBL gene is harbored on a plasmid given that contigs will break at points where repeat regions collapse (i.e., MGEs that are often upstream/downstream of AMR genes).

Reviewers' comments:

Reviewer #1 (Remarks to the Author):

The manuscript describes the use of whole-genome sequencing (WGS) to add value to Japan's national AMR resistance surveillance system (JANIS) in a retrospective study. The study shows how genomics surveillance can clarify antimicrobial resistance mechanisms and provide a complete classification of 3rd generation cephalosporins. It also uses systematic long read sequencing to characterize representative plasmids outside of previous reference country plasmids and allowed their geographical distribution and highlight potential plasmid transfer events.

The manuscript suggests the generation of a blueprint for the integration of genomic epidemiology into a laboratory-based surveillance system in a retrospective study at the National Reference Laboratory (NRL) level, but the description of how this is sustainable through time is not clearly stated, neither how the genomic analysis is integrated at the NRL. Considering the title, this description should be further developed.

Response: We substantially modified the second last paragraph in the Discussion as follows:

“Our genomic surveillance (the Japan Antimicrobial Resistant Bacterial Surveillance, JARBS) was designed as a research project to be linked to the established national surveillance program, “Japan Nosocomial Infections Surveillance (JANIS)”¹⁶. JANIS is funded by the Ministry of Health, Labour, and Welfare (MHLW) and managed by the National Institute of Infectious Diseases. It collects all routine bacterial culturing and antimicrobial susceptibility testing results, including both culture-positive and culture-negative data of all sample types. Hospitals participating in JARBS were recruited from those participating in JANIS, and each strain collected in the hospitals had specimen IDs linked to the JANIS database. The genomic analysis of strains collected in JARBS is integrated at the National Institute of Infectious Diseases, thereby complementing the comprehensive phenotypic surveillance performed by JANIS, and enabling data linkage between the two databases. To ensure the sustainability of the integrated genomic analysis over time, we have made continuous efforts to secure financial support from the MHLW as stated in the new national action plan on AMR starting in 2023. Moreover, we provide feedback reports to each participating hospital, informing them about the presence or absence of major antimicrobial resistant genes in their respective strains. This serves as an incentive for hospitals to maintain their participation in genomic surveillance. Furthermore, we initiated the second phase of JARBS in March 2023.”

(Page 17, Lines 426-435)

This manuscript add value in comparison to other publications describing the use of WGS in AMR surveillance (1, 2), particularly with a dataset of great quality considering that is well phenotypical characterized and confirmed at the NRL with further short reads and long reads study.

The methodology is perfectly described and all software used are publicly available which is essential for reproducibility.

Response: We appreciate your positive feedback.

In line 255 where Figure 6a is mentioned as a Sankey plot reveals that most strains carrying blaIMP-6 had IMC values ≤ 2 ug/ml, this is not clear, whereas is stated in Supplementary figure 20.

Response: We apologize for this typographical error. The text has been revised from “most strains” to “all strains.” In addition, we have added the number of strains in parentheses after each carbapenemase gene in Figure 6, as in Supplementary Figure 20.

In conclusion, the manuscript is clear and accessible, with good and complete references and reproducible and access data. The abstract perfectly summarize the content of the work. It would add original value to this work the expanded description of integration of WGS in a blueprint of a very well established AMR national surveillance system with systematic short and long read data and provide a dataset linked to other data sources for further studies.

Response: We truly appreciate your positive comments.

Reviewer #2 (Remarks to the Author):

Kayama et al. provide an extensive molecular epidemiological study of third generation cephalosporin (3GC) Enterobacterales with reduced meropenem susceptibility in Japan. The authors importantly were able to confirm narrow-spectrum beta-lactam, beta-lactam/beta-lactamase inhibitor (BL/BLI), 3GC and carbapenem AST phenotypes with 4195 sequenced Enterobacterales species providing a robust analyses of AMR gene carriage and associations with particular clinically relevant AST phenotypes as well as plasmid vectors.

These analyses are critical to understand genotype/phenotype correlations across multiple geographical regions. Crucially, highlighting concordance and importantly discordance with susceptibility tested versus AMR gene detection is important to inform potential molecular diagnostic measures. This is illustrated by the fact that 15% of carbapenemase positive isolates had carbapenem susceptible profiles (Fig 6 and Fig. S20). While these findings are well documented throughout the manuscript, I believe that a concordance measure of AMR gene vs. AST phenotype (e.g., Kendall's W) would be a great quantification and presentation of the results that are shown primarily through figures that are often difficult to assess on their own. I think this could simply be focused on some of the important antibiotic classes, such as carbapenemases as well as BL/BLIs to name a few.

Response: We have added the quantification using Kendall's W for 3GC, carbapenems, and ABPC/SBT as follows:

Regarding 3GC:

"Moreover, the Kendall's coefficient of concordance (W) between carriage of an ESBL gene and 3GC resistance was 0.55 among the 4,088 3GC-resistant and 107 3GC-susceptible strains." (Page 4, Lines 90-92)

"Kendall's W between carriage of an ESBL or AmpC gene and 3GC resistance was 0.58" (Page 9, Line 213)

Regarding carbapenems:

"Overall, Kendall's W between the carriage of a carbapenemase gene and meropenem-resistant/intermediate/susceptible phenotype was 0.85 among the 4088 strains resistant to 3GCs." (Page 12, Lines 283-285)

Regarding ABPC/SBT:

"Kendall's W between the *bla*_{TEM-1} gene co-occurrence and ABPC/SBT resistant/intermediate/susceptible phenotype was 0.74 among strains harboring *bla*_{CTX-M-15} or *bla*_{CTX-M-27}. However, the concordance was lower among strains harboring *bla*_{CTX-M-14} (Kendall's W = 0.57), suggesting the presence of another unknown factor. This is reflected in that 39% of strains non-susceptible to ABPC/SBT did not carry *bla*_{TEM-1} (top left in Supplementary Figure 10)." (Page 8, Lines 189-196)

While perhaps some figure legends could also include additional detail, most of the

figures provide nice visual aids to make sense of particular AMR gene relationships.

Response: We appreciate your positive feedback. To supplement the information, we have added the following to the Figure 6 and Supplementary Figure 10 legends:

“The numbers in the parentheses indicate the number of strains carrying the gene (at the left) and showing the MIC (at the right).”

Additionally, the methods are well described with great supporting metadata as they mention could be of great use to persons using publicly available data to build prediction models. Below are some further comments that could further aid in clarification/interpretation as well as minor editing suggestions.

Response: We truly appreciate your positive comments. We have revised the text according to the following suggestions.

Major Comments:

Line 104 – 148: There are inherent limitations to inferring ESBL encoding gene genomic context with reference-based alignments that should be considered. While the co-occurrence of ESBL presence with a certain ID/COV% of a select few reference plasmids does suggest the potential of each respective plasmid as a vector for CTX-M carriage, these interpretations should only be stated as associations or ‘likely carriage’ unless there is further evidence of co-carriage within each plasmid (i.e., additional long-read sequencing or more extensive S1-PFGE + SB) especially at this large of a scale. If authors want to use a 90% cutoff threshold to determine ‘likely carriage’, at the least, inferring genomic context with short-read alignments should be mentioned as a limitation. This has been addressed in part in line 383 – 390, but should mention that this threshold cutoff does not rule out potential for other ESBL encoding gene genomic contexts.

Response: We have addressed this point by modifying the text as follows:

“Hence, the *bla*_{CTX-M-8} gene showed associations with plasmids that were highly similar to the reference complete plasmids, although this threshold cutoff does not rule out the potential for other ESBL-encoding gene genomic contexts.” (Page 6, Lines 140-142)

Line 171-179: While the network graph in Fig. S10 does a decent job of visually showing nodal sharing between *bla*TEM-1b and *bla*CTX-M-14/*bla*CTX-M-15 in contrast to *bla*CTX-M-27, it may be useful to present *bla*TEM-1b presence/absence data in a complimentary table (with co-occurrence of each respective *bla*CTX-M variant) to

demonstrate the relationship with SAM susceptibility.

Response: We have added a new Supplementary Table 2 that presents the blaTEM-1b presence/absence data with the co-occurrence of each respective blaCTX-M variant. Examination of the table demonstrated a need to modify the main text as follows:

“A main factor underlying the large difference was the frequency of *bla*_{TEM-1} gene co-occurrence (Supplementary Table 2 and Supplementary Figure 10), which is known to cause resistance to ABPC/SBT¹⁹. Kendall’s W between the *bla*_{TEM-1} gene co-occurrence and ABPC/SBT resistant/intermediate/susceptible phenotype was 0.74 among strains harboring *bla*_{CTX-M-15} or *bla*_{CTX-M-27}. However, the concordance was lower among strains harboring *bla*_{CTX-M-14} (Kendall’s W = 0.57), suggesting the presence of another unknown factor. This is reflected in that 39% of strains non-susceptible to ABPC/SBT did not carry *bla*_{TEM-1} (top left in Supplementary Figure 10).” (Page 8, Lines 189-196)

Line 237 – 243: The difference in FEP susceptibilities doesn’t indicate that *K. pneumoniae* (Kp) co-carried an ESBL gene with blaDHA-1. I would report the number of isolates that carry both ESBL + blaDHA-1 29/74 (39%) in Kp as reported in Fig. S11 and how many are concordant with predicted phenotypes as shown in Fig. S18.

Response: We have modified the text to address this point as follows:

“Among the 397 3GC-resistant *E. coli* strains carrying the AmpC β-lactamase genes, 26 (7%) also possessed an ESBL gene (Supplementary Figure 11b). In contrast, among 74 3GC-resistant *K. pneumoniae* strains carrying the AmpC β-lactamase genes, 29 strains (39%) possessed an ESBL gene (Supplementary Figure 11c), which was significantly higher than that of *E. coli* ($p < 10^{-13}$, chi-square test). Antimicrobial susceptibility profiles of the 3GC-resistant *E. coli* and *K. pneumoniae* strains carrying the AmpC β-lactamase genes are shown in Supplementary Figure S18, stratified by *bla*_{CMY-2} and *bla*_{DHA-1}. As a result, 92% and 91% of the *E. coli* strains carrying *bla*_{CMY-2} and *bla*_{DHA-1} were concordant with the predicted phenotype (susceptible to cefepime), while 66% of the *K. pneumoniae* strains carrying *bla*_{CMY-2} exhibited concordance with it, likely due to the increased proportion of strains also possessing an ESBL gene.” (Page 11, Lines 257-268).

Additionally, Figure S18 was modified slightly to include only strains resistant to 3GCs.

Minor Comments:

Line 90: May be helpful to include frequency counts in supplemental table 1 for each SHV, TEM, GES variant, otherwise not sure how helpful the inclusion of this table is to the manuscript.

Response: We have added frequency counts to Supplementary Table 1, accordingly.

Line 97 – 99: Was there any assessment of likely AmpC derepression through assessment of qRT-PCR and/or ampC promoter mutations? The assessment of ampC promoter mutations using AMRFinder plus could potentially help clarify the genetic mechanisms.

Response: We have added an assessment of ampC promoter mutations using AMRFinder plus, and the corresponding results to the revised manuscript, as follows: “We also confirmed that the five *E. coli* strains had mutations in the promoter region of the chromosomal AmpC (*bla_{EC}*) gene, which can increase gene expression. Among the total 545 *E. coli* strains with cefepime MIC ≤ 2 $\mu\text{g/mL}$, 63% (341) carried five types of mutations in the promoter region (C-11T, C-42T, G-15GG, T-14TGT, T-32A), as detected by AMRFinderPlus¹³. The proportion was significantly higher ($P = 0.008$, Fisher’s exact test) than 20% (2 of 10) of *E. coli* strains with cefepime MIC > 2 $\mu\text{g/mL}$, suggesting that AmpC enzyme is not produced at high levels. Meanwhile, the remaining 37% of the 545 *E. coli* strains with cefepime MIC ≤ 2 $\mu\text{g/mL}$ likely exhibited other mechanisms that were undetectable by AMRFinderPlus, for example the incorporation of IS10 and IS911 into the promoter region¹⁴.” (Page 5, Lines 103-111)

Line 165; 168: Should spell out levofloxacin and ciprofloxacin

Response: We have spelled out levofloxacin and ciprofloxacin, accordingly (Page 8, Lines 179, 182, 183).

Line 187: Would clarify these are plasmid encoded AmpC genes

Response: We have addressed this point by adding the following text to the manuscript: “The 1041 strains possess plasmid-encoded and chromosomal AmpC genes, as 53% of them were identified as *Enterobacter*, *Citrobacter*, *Morganella morganii*, or *Klebsiella aerogenes*, which are known to harbor chromosomal AmpC genes^{21,22}.” (Page 9, Lines 207-209)

Line 201: Should be IncFII

Response: We have fixed it to be IncFII. (Page 9, Line 222)

Line 224: The detection of plasmid replicon types enriched in particular hospitals does not suggest an outbreak. Sequence similarity between isolates harboring such plasmids would need to be included with additional temporal/spatial epidemiological data.

Response: Given the absence of additional temporal/spatial epidemiological data, we

have deleted the description regarding potential outbreaks. (Page 10, Line 245)

Line 248: Would clarify how percentages are being calculated, i.e., the percent of total carbapenemase

Response: Yes, this refers to the percentage of the total number of carbapenemase genes detected in the present study ($N=307$, as shown in Supplementary Figure 19). We have added “out of the total number of carbapenemase genes; Supplementary Figure 19” to the sentence. We have also fixed minor typos regarding the percentages in the sentence.

Line 270: I believe that “AMK” is supposed to be “Augmentin – amoxicillin-clavulanate”. However, it is commonly used as an abbreviation for amikacin, an aminoglycoside antibiotic, so would be careful with nomenclature. In the past, I have used the abbreviation “AMC” to avoid this confusion.

Response: We carefully checked the nomenclature, and have confirmed that the American Society of Microbiology (ASM) uses “AMK” for amikacin, as shown in Supplementary Table 4 and the “Antibacterial agents” section in <https://journals.asm.org/abbreviations-conventions>. Moreover, the ASM uses “AMC” for amoxicillin-clavulanic acid.

Line 427: Did non-*E. coli*/*K. Pneumoniae* Enterobacterales species have different selection criteria, i.e., 3GC AST was not performed, rather reduced MEM susceptibility was inclusion criteria? If so, is there a reason why or is this due to how isolates are surveilled through JARBS?

Response: Yes, non-*E. coli*/*K. Pneumoniae* Enterobacterales species were collected using only the epidemiological cutoff for screening carbapenemase. This is due to how isolates are surveilled through JARBS.

Line 532: To elaborate on previous point, progressive Mauve alignments of long- to short-read assemblies will still not definitively determine for a proportion of assemblies whether a colinear block that carries an ESBL gene is harbored on a plasmid given that contigs will break at points where repeat regions collapse (i.e., MGEs that are often upstream/downstream of AMR genes).

Response: We added the following description:

“This genome alignment approach using progressive Mauve does not exclude the possibility of contigs carrying an AMR gene breaking at points where repeat regions collapse due to mobile genetic elements.” (Page 23, Lines 582-584)

REVIEWERS' COMMENTS

Reviewer #2 (Remarks to the Author):

I have carefully reviewed the responses to my critiques and believe the authors have adequately addressed the issues to my satisfaction. There is no need for me to review the manuscript any further.